# Employing Extracellular Matrix-Based Tissue Engineering Strategies for Age-Dependent Tissue Degenerations

**DOI:** 10.3390/ijms22179367

**Published:** 2021-08-29

**Authors:** Yeonggwon Jo, Seung Hyeon Hwang, Jinah Jang

**Affiliations:** 1School of Interdisciplinary Bioscience and Bioengineering, Pohang University of Science and Technology (POSTECH), Pohang 37673, Korea; j920901@postech.ac.kr; 2Department of Mechanical Engineering, Pohang University of Science and Technology (POSTECH), Pohang 37673, Korea; shhwang725@postech.ac.kr; 3Department of Creative IT Engineering, Pohang University of Science and Technology (POSTECH), Pohang 37673, Korea; 4Institute for Convergence Research and Education in Advanced Technology, Yonsei University, Seoul 03722, Korea

**Keywords:** extracellular matrix, aging, tissue dysfunction, age associated diseases, biomaterial, cell delivery, tissue engineering, 3D bioprinting, regenerative medicine

## Abstract

Tissues and organs are not composed of solely cellular components; instead, they converge with an extracellular matrix (ECM). The composition and function of the ECM differ depending on tissue types. The ECM provides a microenvironment that is essential for cellular functionality and regulation. However, during aging, the ECM undergoes significant changes along with the cellular components. The ECM constituents are over- or down-expressed, degraded, and deformed in senescence cells. ECM aging contributes to tissue dysfunction and failure of stem cell maintenance. Aging is the primary risk factor for prevalent diseases, and ECM aging is directly or indirectly correlated to it. Hence, rejuvenation strategies are necessitated to treat various age-associated symptoms. Recent rejuvenation strategies focus on the ECM as the basic biomaterial for regenerative therapies, such as tissue engineering. Modified and decellularized ECMs can be used to substitute aged ECMs and cell niches for culturing engineered tissues. Various tissue engineering approaches, including three-dimensional bioprinting, enable cell delivery and the fabrication of transplantable engineered tissues by employing ECM-based biomaterials.

## 1. Introduction

The extracellular matrix (ECM) is an essential non-cellular component of the body that is present in all tissues and organs [1]. The ECM organizes large fibrillar three-dimensional (3D) networks composed of glycosaminoglycans (GAGs) and various types of proteins such as collagen, fibronectin, elastin, and laminin. Among the numerous constituents of the ECM, fibrous proteins (i.e., collagen and elastin), adhesive glycoproteins (fibronectin and laminin), and glycosaminoglycans (i.e., hyaluronic acid) are recognized as the main components [2]. Although the compositions differ based on the type of tissue, type I and II collagen are the primary constituents and are associated with other ECM constituents [3]. The ECM offers biophysical, biochemical, and biomechanical cues for cellular components. Furthermore, it provides structural support, functions as an adhesive substrate, presents growth factors to its receptors, sequesters, stores growth factors, senses, and transduces mechanical signals [4,5]. Furthermore, the ECM is involved in regulating several cellular functions, such as survival, adhesion, migration, proliferation, differentiation, and supporting cells for binding [4,6].

The ECM undergoes changes in the aging cell. The ECM is degraded and deformed, whereas the expression of ECM components is up or downregulated [7]. As the ECM provides a microenvironment for cells, alterations in the ECM may affect cellular functionalities. Abnormal changes in the ECM contribute to alterations in pathological conditions, and stem or progenitor cells fail to proceed with normal regeneration [8]. Hence, regeneration strategies must be applied to overcome age-associated dysfunctions and disorders. Currently, ECM-based biomaterials are prominently utilized in regenerative therapies owing to their high biocompatibility and biomimetic properties. ECM-based biomaterials can be employed in relatively simple applications (i.e., injectable fillers) to complicate biofabrication methods (i.e., bioink). Owing to the injectability and biomimetic characteristics of ECM-based materials, more detailed biomimicking engineering is now feasible. Among the latest biofabrication technologies, 3D bioprinting technology that adopts biomaterials as bioinks has been highlighted [9].

Herein, we describe the aging and rejuvenation within the framework of ECM. The functions of the ECM components, changes in the ECM with aging, age-related disorders, and rejuvenating strategies are introduced in the following sections. The ECM is mentioned in every section of this article to emphasize its importance in both understanding the aging process and developing regeneration strategies. We discuss current rejuvenating approaches using ECM-based biomaterials and biofabrication methods used in the past five years to facilitate the further development of ECM-based materials such that issues in the field of tissue engineering can be solved.

## 2. Changes in ECM Components with Aging

Tissues and organs undergo physical and physiological changes during aging, and the aging process varies depending on the tissue and its components. Not only cellular components but also non-cellular components such as ECM [8,10] undergo aging (Table 1). The ECM undergoes constant regulation through deposition, modification, and degradation, primarily via matrix metalloproteinases (MMPs) in normal tissues. However, at the senescence stage, changes occur in both the ECM components and enzymes. The roles and alterations of some abundant and important ECM components, including collagen, are partially known; however, not every ECM component has been investigated in terms of their specific functions and aging process [7].

### 2.1. Collagen

Collagen is one of the most abundant proteins in the human body. More than 28 different types of collagen exist, and the ratio varies for different tissues. Collagen primarily provides structural properties and resilience to tissues. Typically, collagen constitutes a significant portion of the ECM, and in the dermis, collagen is a major component, constituting up to 75% of the total composition [11]. However, aging cells fail to regulate collagen synthesis, resulting in changes in the amount and ratio of collagen. Aging modulates the expression of collagen differently, and several types of collagen in various tissues are down- or upregulated depending on the circumstances [12,13,14]. In some cases, collagen is overexpressed with aging, inducing fibrosis [15]. It has been reported the expression of collagen IV increased in age-related neural disorders [16]. Meanwhile, the downregulation of collagen due to degradation has been observed in the senescence path [17]. Additionally, collagens undergo modifications (i.e., mineralization and accumulation of advanced glycation end-products), and these modifications affect the susceptibility of collagens to MMP-mediated degradation [18].

### 2.2. Fibronectin

Fibronectin is an abundant glycoprotein that is essential for ECM assembly. Fibronectin can bind various ECM molecules such as collagen I and III, gelatin, thrombospondin, and heparin [19]. Fibronectin typically functions through interactions with the ECM network. Fibronectin mediates cellular activities by binding to integrin or syndecan receptors on the cell surface [20]. The gene expression of fibronectin is downregulated with collagen in senescent hepatic stellate cells [21]. Meanwhile, fibronectin expression increases in aortic endothelial cells in vivo with the age of donors [22]. Furthermore, the expression of fibronectin increases in patients with Alzheimer’s disease [16]. Increased fibronectin mRNA accumulation has been observed in in vitro cellular senescence [23]. Both the downregulation and upregulation of fibronectin were indicated in senescent cells.

### 2.3. Elastin

Elastin is a fibrous self-assembling protein that contributes significantly to the mechanical strength, elasticity, firmness, and suppleness of tissues. Elastin facilitates the appropriate functioning of organs and is present in almost all connective tissues [24]. The primary age-dependent change in elastin is degradation. The expression levels of elastin-degrading enzymes (i.e., cathepsins and elastase) increase with aging [25,26]. Age-related destruction of the elastin fiber structure has been investigated in the skin and artery [25,27]. It was reported that aging affects the structural and mechanical properties of elastic fibers, resulting in the fragmentation and structural reorganization of elastin [27,28].

### 2.4. Laminin

Laminin is a large heterotrimeric glycoprotein that is integral to the basement membrane. Laminin is a major component of the basal lamina; it is one of the layers of the basement membrane and contributes significantly to the lamina’s structure. Laminins aid mediating processes such as cell differentiation, proliferation, adhesion, migration, and tissue maintenance based on receptor- and matrix-binding properties [29]. Some laminins have been reported to be upregulated in some aging tissues, particularly in diabetic basement membranes [30]. A recent study demonstrated that decreased laminin production is an early indicator of an aging lung [31]. Another recent study reported that the loss of laminin-α4 results in non-progressive impaired neurotransmissions and premature morphological alterations typically associated with an aging neuromuscular junction [32].

### 2.5. Hyaluronic Acid

Hyaluronic acid (HA) is a substantial GAG component that comprises glucuronic acid and hyaluronan. HA exhibits the unique characteristic of water binding, thereby allowing it to preserve a large amount of moisture, which is important for tissue hydration [33,34]. HA contributes significantly to the structure and organization of the ECM. However, during aging, the elasticity, turgidity, and mechanical strength of HA deteriorate [35]. Age-induced deterioration, which is relevant to symptoms of aged skin such as wrinkles, has been observed in the skin [34]. The decline in both the total mass and polymer size was investigated in aging skin [36]. Furthermore, the water-binding capacity decreased as senescence progressed [37]. The level of HA decreased in the lamina propria with age [38].

## 3. Influence of Age-Related Changes of ECM on Tissues

Changes in the ECM during aging not only result in cellular senescence but also affect tissue dysfunction. Aging is characterized by reduced tissue function and regenerative capacity [39]. During cellular senescence, the ECM and ECM-related components undergo abnormal regulation and remodeling. The mechanisms underlying these abnormalities and cell behavior have not yet been elucidated at the molecular level; however, it is certain that such defect changes in the ECM dynamics affect cell and tissue functions [40]. Age-related changes in the ECM mediate cell proliferation, differentiation, inflammation, and apoptosis. Furthermore, it affects the recruitment, differentiation, and functional integration of stem- and tissue-specific progenitor cells [8]. As the strict regulation of ECM synthesis and remodeling is directly associated with healthy tissue conditions, alterations in ECM remodeling can affect the course and progression of several other pathological conditions, including fibrosis, disorders, and cancers [2]. A recent study revealed that the ECM compositions of young, adult, and aged tissues are different and that the ECM age significantly affects the proliferative ability, maturity, and stress response of cells. Aging cues, which accelerate the aging process, are presented from aged ECM on an in vitro test platform [41]. The major alterations in tissues related to the ECM during aging, which are described in detail in the subsections below, are mass loss, moisture loss, fibrosis, and dysfunction.

### 3.1. Mass Loss

The regenerative function of most tissues declines with age, and mass loss occurs in various tissues. Although the correlation between ECM alteration and tissue loss is yet to be elucidated, it is certain that the ECM comprises tissues and may likely contribute to deviations. As the ECM promotes cell proliferation and adhesion, structural and biochemical changes in the ECM may affect cell maintenance [42]. The number of proliferating cells and the number of newly synthesized ECMs typically decrease during aging, resulting in mass loss. In particular, certain muscle tissues indicate a considerable decline associated with aging, resulting in interference with physical activities. Additionally, brain tissues decline gradually with aging, and this tendency is consistent with the decline in cognitive performance among the elderly [43]. ECM alterations induce neural cell death, and the decline of neurons may result in ECM degradation caused by the proteolytic activity of MMPs [44]. Dermal collagen fibrils undergo gradual loss and fragmentation. Alterations and reduced production of collagen at the dermal site impair the structural integrity and induce thin skin, rendering the skin more vulnerable with increased age [45,46].

### 3.2. Moisture Loss

Connective tissues, such as ligaments and tendons, are innately stiffened with aging. This is the result of age-related water content loss and ECM turnover. During aging, the mechanical properties and solubility of collagen diminish, thereby increasing mechanical stiffness; consequently, joint motion decreases and pain increases [47]. Proteoglycans, one of the major components of the ECM, maintain the hydration of tissues and the viscoelasticity of the disk to support mechanical loads. Nucleus pulposus, the core of the disk, receives water molecules through the effect of chondroitin sulfate proteoglycans. However, as the disk ages, the composition of the ECM shifts from proteoglycan-rich to fibrotic collagen-rich. Consequently, the water-binding ability degenerates, and the disc becomes stiffer [48]. Nevertheless, distinct from water content matter, the ECM constitutes 90% of the dry weight of the cartilage; hence, pathological changes in the ECM are regarded as key for understanding cartilage disorders [49]. Additionally, skin aging is associated with moisture loss. HA, which is abundant in the skin, loses its capacity to retain water with aging; eventually, the skin becomes dry and lusterless [37].

### 3.3. Fibrosis

Fibrosis is the excess of fibrous tissues, and sclerosis is the stiffening of tissues. Both pathological states are typically caused by the deposition of connective tissues with abundant collagen and glycosaminoglycans. They are indicators of aging and are related to ECM alterations. Aging promotes the structural remodeling of the ECM, resulting in fibrosis and tissue function deterioration. Fibrosis is primarily caused by increased concentrations of collagen fibers, which results in the development of fibrotic conditions. Collagen deposition in the heart contributes to cardiac fibrosis, which eventually decreases cardiac function [50]. Additionally, vascular stiffening is correlated with fibrosis. Arterial stiffening is primarily caused by the alteration of ECM components during aging [51]. As we age, elastin fibers become weaker, and stiff collagen fibers are burdensome. Unlike elastin, the concentration of collagen increases with age; this causes fibrosis, which increases arterial stiffness significantly and induces cardiovascular diseases [52]. Hepatic fibrosis is recognized as ECM dyshomeostasis, which is caused by ECM aging. Fibrillar ECMs such as collagens accumulate in the liver and can progress to cirrhosis or worse [53]. Idiopathic pulmonary fibrosis, an age-related lung disorder, occurs primarily in the elderly population. Lung fibrosis occurring in the elderly is characterized by increased type I and III collagen, elastin, and fibronectin [54].

### 3.4. Tissue Dysfunctions

Many disorders and tissue dysfunctions are assumed to be associated with aging. They occur because of epigenetic changes, protein deformation, and many other factors. Apart from mass loss and fibrosis, several disorders, including diabetes and Alzheimer’s disease, are associated with aging [55]. A recent study showed that elastin in the brain might induce pathological changes in Alzheimer’s disease, and elastin is known to be fragmented and released with aging [56]. Significant ECM changes, including increased expression of collagen IV and fibronectin, are presumed to affect disease progression in patients with Alzheimer’s disease [16]. The glycemic condition caused by diabetes and aging can affect the ECM constituents and result in complications [57]. Although not many studies of age-related diseases investigated the direct relationship with ECM aging, some studies have reported that ECM aging and age-related tissue dysfunctions are indirectly linked. The pathogenesis of diabetic kidney disease is characterized by increased ECM accumulation, which causes glomerular thickening and tubulointerstitial fibrosis. Furthermore, the increased levels of MMPs are correlated with the development and progression of diabetic nephropathy [58]. The glycation of proteins that leads to the formation of advanced glycation end-products (AGEs) is also known to be mainly related to aging [59]. The modification of proteins (e.g., collagen and fibronectin) by AGEs may contribute to an alteration in their structure and function, indicating the consequent progression of several diseases, such as diabetic nephropathy, retinopathy, atherosclerosis, arthritis, cardiovascular diseases, and neurodegenerative diseases [60].

## 4. Utilization of ECM for Regeneration Strategies

The senescence of the ECM affects tissue dysfunction and tissue regeneration failure. To rejuvenate aged tissues and restore their functions, regenerative strategies are indispensable. Tissue engineering, which focuses on restoring, maintaining, and improving the damaged tissues, is a possible treatment. Tissue engineering is an interdisciplinary field in which cells and matrices are applied. The cell source can be varied from primary lineages to stem cells (SCs), and synthetic polymers or the ECM can be employed as a matrix [61]. Synthetic polymers such as polycaprolactone (PCL), polylactic acid (PLA), and polyvinyl alcohol (PVA) have been typically used in early studies. However, owing to the hydrophobicity and non-natural residues of most synthetic polymers, naturally occurring biomaterials such as ECM-based materials and some hybrid biomaterials have been extensively investigated for biomedical applications [62]. Furthermore, numerous recent 3D tissue engineering strategies have been actively researched to treat age-related tissue dysfunctions by employing ECM-based materials [63,64,65]. In this section, the types of ECM-based materials and their applications as tissue engineering strategies are introduced for the regeneration of age-dependent tissue dysfunctions.

### 4.1. Types of ECM-Based Materials for Biomedical Applications

As aged ECMs adversely affect cellular regulations and appear in fewer numbers, replacing them with ECM-based materials is an option. ECM is a typical biomaterial that can be used for in situ tissue regeneration. This biomaterial can interact with the microenvironment and induce in situ tissue regeneration [66]. Additionally, ECM-based materials are used as an injectable soft tissue filler for filling up the volume lost from ECM degradation during aging. Many commercial filler products based on ECM components can be used in clinical applications [67]. At the dermal site, in particular, the number of ECM components decreases on the outside. Aging features such as wrinkles and loss of skin elasticity can be rejuvenated through antiaging approaches, including dermal filling [68]. Soft tissue fillers based on ECM constituents have been reported to affect tissue regeneration [69,70]. ECM-based materials can be processed from native tissues to products through isolation, decellularization, and purification (Figure 1) [71]. Native collagen, fibronectin, elastin, and the like are commercially available. They are typically isolated from specific ECM-enriched tissues, such as collagen from the dermis and fibronectin from the plasma. Purified ECM components are used for various applications, from biomedical to cosmetics, and have demonstrated potential for therapeutic use [72,73,74,75,76].

Collagen is frequently used as a basic biomaterial because it is present in high amounts in the ECM. In some cases, the ECM is modified to enhance its unique characteristics or to eliminate its harmful effects, and atelocollagen is one of the most popular modified ECMs [76]. By removing telopeptide from collagen to form atelocollagen, the chance of an immune reaction is reduced. Protocols to produce atelocollagen with high purity at a faster speed have been developed in some studies [77]. In other studies, atelocollagen was used to treat tendons and periodontal tissues [78,79]. Additionally, some examples of improvement in collagen functions have been reported, including the enhancement in the ability of collagen to promote cell proliferation and anti-thrombogenic activity via conjugation with chondroitin sulfate. Although collagen is known to induce thrombosis, chondroitin-sulfate-modified collagen may weaken the induction [80]. Apart from modification, a novel scaffold can be achieved by obtaining an anisotropic structure or inducing the orientation of collagen fibers. The aligned structure of collagen provides insights into neural and vascular endothelial cells [81,82].

HA is an injectable hydrogel that is vital to hydration and the space-filling capacity [37]. However, HA exhibits relatively weak mechanical properties, and it dissolves and degrades rapidly. Hence, HA has been modified using numerous methods for biomedical applications, such as controlling the molecular size or viscoelasticity [37,84]. Recently, novel methods for modifying HA-based hydrogels to achieve enhanced stiffness and control degradability while sustaining injectability have been reported. The combination of dynamic covalent crosslinking with thermoresponsive engineered proteins enhanced mechanical stiffness, and the crosslinking of monoaldehyde-modified HA with carbohydrazide-modified gelatin decelerated the degradation [85,86]. Skin rejuvenation and joint treatment are representative applications of HA that involve the injectability of HA [87,88]. HA is introduced in joints via either an injectable hydrogel or solid scaffold form to relieve inflammation and repair the cartilage [89]. Enhancing the water-binding capability of HA is another example of a modification that may aid tissue hydration. A modified interpenetrating polymer network that entraps HA and water, thereby aiding moisturization, has been reported [90].

To apply a multicomponent matrix for the microenvironment, the decellularized ECM (dECM) is typically employed for biomedical applications. The dECM is created by eliminating cellular components from the native tissue. Tissue-specific ECMs are important for the development and functioning of normal tissues, and the dECM is a representative example [91]. Through decellularization, only bioactive ECM components that do not harm the immune system remain and have a similar ratio of ECM components as the native tissues [92]. A decellularizing protocol was developed to maintain the delicate structure and composition of the native tissue. Hence, the dECM was compared to the native tissue using proteomics and protein staining imaging [93]. Furthermore, the dECM can be classified based on the source of the original tissues. Using only specific tissues as the source of the dECM, tissue-specific compositions can be provided [91,94,95]. The dECM-based hydrogel can be applied as a bioink for 3D bioprinting, injectable hydrogels, cell delivery, and translational medicine [96]. However, dECM lacks mechanical strength and stability. Post-decellularization processes (e.g., hybridization of dECM with a crosslinking agent) have been developed to enhance the mechanical properties of dECM [97]. A recent study introduced the light-activated dECM, which can be polymerized via a dityrosine-based crosslinking system, to improve the inferior mechanical properties of the dECM [98]. In the case of cartilage tissue, resistance to compressive forces is required. Synthetic polymers are often hybridized with dECM using crosslinking agents to satisfy the physical properties of cartilage tissues. The hybridization of dECM with cell-encapsulated microparticles or platelet-rich plasma (PRP) is regarded as a post-decellularization process [97].

### 4.2. ECM-Based Materials as Cell Delivery Carriers

Because the surrounding of an aged ECM is damaged, replacing only the ECM might not be sufficient to regenerate the aged tissues. Hence, cell delivery may be required for successful tissue regeneration. However, there are limitations to cell delivery; thus, tissue engineering techniques are being studied to supplement the insufficient capacity of past cell delivery systems [99]. Tissue engineering is a complex process that involves recellularizing the ECM matrix. Recellularization requires a specific environment that provides biochemical and biomechanical conditions similar to those of a specific organ [97]. Similarly, niches that provide a microenvironment for the delivered cells are essential to design effective cell carriers [64]. For instance, type V collagen has been shown to induce the formation of human islet organoids and the generation of endocrine cells by profiling extracellular proteins in the pancreatic ECM. Type V collagen is an indispensable factor in the recapitulation of niches for pancreatic cells [100]. Essential factors regulating cell functions, cell survival, and cell retention must be considered when deciding the materials for cell delivery [101]. Many studies have revealed that ECM-based biomaterials showed improved results in cell delivery and successfully regenerated defective tissues. Therapeutic cell delivery restored the functions of organs such as the heart, nerve, joint, thyroid, and the like using cells and ECM-based biomaterials [102,103,104,105]. The cell-delivering material replenishes the volume first; subsequently, the delivered cells secrete matrix components, thereby restoring the regeneration capacity [106]. Generally, progenitor cells, such as embryonic SCs, induced pluripotent SCs, and mesenchymal stem cells (MSCs), are combined with ECM-based biomaterials for cell therapeutics [107]. MSCs showed improved potency when expanded on the dECM [108]. Moreover, ECM-based hydrogels may enable human organoids to be delivered in vivo for clinical purposes [109]. Although current clinical trials based on SCs are still at an early stage, several studies have revealed that SC-based approaches are potential treatments for degenerative diseases, such as Parkinson’s disease and osteoarthritis [110,111].

Cardiovascular diseases are the leading cause of death and are of high risk in the elderly. To prevent fatal diseases, cell delivery for the cardiac disease has been actively investigated to restore tissue function and cardiac regeneration [112]. The application of ECM-based biomaterials has resulted in successful cardio cell delivery [113]. A recent study validated that the combination of dual SCs and ECM-enhanced cell retention, engraftment, and maturation. The result of cell delivery showed significantly higher gene expression, indicating better integration with the host myocardium [114]. MSCs were delivered in patch form to improve cell adhesion and effectively enhance cardiac remodeling [115]. The development of a heart decellularization method for retaining ECM-derived biochemical cues and promoting angiogenesis has been attempted [116].

Understanding neural-specific ECMs is crucial for the treatment of neurodegenerative diseases because the neural extracellular matrix is known to differ from the normal ECM in other organs. The neural ECM has a low fibrous protein content and high glycoprotein and carbohydrate contents. One of the key carbohydrate components in the brain ECM is chondroitin sulfate proteoglycans. It is found that chondroitin sulfate GAG matrices promote neural stem cell maintenance and efficacy [117]. Proteomic analysis showed sufficient chondroitin sulfate proteoglycans in decellularized nerve tissues, indicating its suitability for cell therapy applications. Moreover, a recent study established the genipin and EDC crosslinking of dECM to enhance stability [118]. Therefore, some studies have optimized the neural microenvironment using the dECM. dECM made from brain tissue accelerated the neural network formation in vitro [119]. The use of laminin enriched ECM hydrogel enhanced the retention rate of the delivered cells. Because of the importance of laminin in the neural ECM, laminin-derived peptides have been applied to promote neural cell adhesion in neural progenitor cell delivery research [120].

Osteoarthritis is another age-related joint disease in which cell delivery therapy has progressed. The cartilage is composed of specialized cells known as chondrocytes, which produce a large number of collagenous extracellular matrices, and osteoarthritis is a degenerative cartilage disorder [121]. Adipose-derived SCs and MSCs delivered with the ECM were investigated to secrete the newly synthesized ECM, which can restore the ECM-producing function of cartilage [104,122]. The differentiation of MSCs was promoted by dECM, and the ECM derived from human umbilical vein endothelial cells (HUVECs): MSC co-culture improved the osteogenic and angiogenic potential [123]. For further applications, ECM-polymer hybrid or cross-linkable injectable hydrogels are often used as functional biomaterials because the cartilage must endure high pressures [124]. PVA is typically used in combination with ECM materials to achieve both favorable mechanical properties and biocompatibility [125]. Recently, PLGA (poly(lactic-co-glycolic acid)) has also been used with collagen to prepare synergetic hybrid biomaterials [126]. Moreover, several MSC-based clinical trials have been reported to have an effect on cartilage renewal, and trials are ongoing to improve long-term osteoarthritis [111].

### 4.3. 3D Bioprinting Using ECM-Based Bioinks

Aged ECMs around the delivery site may cause delivered cells to fail initially. To overcome the limitation of cell delivery by mimicking the original tissues, transplantable engineered tissues are being developed to produce substitutes [99]. Tissue development is not achieved via cell culture alone. In fact, the surrounding environment is important, and 3D cell deposition is necessitated. Tissue engineering technology enables the fabrication of various types of engineered organs such as the pancreas, liver, kidney, ovary, bladder, cornea, muscle, skin, vessel, esophagus, and the like using SCs and biomaterials (Figure 2) [127,128,129,130,131,132,133,134,135,136,137]. Although numerous hurdles (i.e., low cell survival, immune response, and insufficient functions) remain to be overcome, many studies have revealed the effectiveness of engineered tissue transplantation as a novel regeneration method [138,139]. Some simple tissues (i.e., artificial epidermis) that are relatively easier to engineer, as well as some vital organs that are in high demand for transplantation, have been developed and are progressing favorably [66,127]. Furthermore, to achieve highly precise engineered tissues, novel biofabrication methods such as 3D bioprinting are being developed actively [140,141,142]. Conventional tissue engineering approaches, such as scaffolding and molding, are limited in their capacity to produce precise tissue constructs. Unlike the former fabrication methods, 3D bioprinting is a rapid and efficient method for constructing elaborate engineered tissues [143]. Three-dimensional bioprinting is a type of additive manufacturing system that allows controlled deposition, i.e., several cells can be deposited at the desired locations to bio-mimic natural tissues using bioinks [144]. A 3D bioprinting system can facilitate complex construction using multiple materials and multiple cells by utilizing multiple nozzles [145]. Tissue-specific geometric structures (i.e., convoluted tubules as well as chamber- and lobule-like structures) constructed via 3D bioprinting demonstrated enhanced function and structural maturity [146,147]. The 3D bioprinting system uses bioinks and biomaterial inks to print the product, and ECM-based materials for bioinks have become the popular choice recently [148].

Numerous stem cell therapy companies are attempting to engineer pancreatic tissues [149]. The pancreas is an organ of both the endocrine and exocrine systems, typically known for its role in controlling blood glucose levels. The pancreas undergoes pathological alterations during aging, including fibrosis, and these changes may result in diabetes mellitus [150]. The specific microenvironment of the ECM is vital to the function of metabolic tissues such as pancreatic islets. Pancreatic cells are expected to function as intended in a specific microenvironment (i.e., collagen type I, IV, and VI abundant), where ECM–cell interactions exist [151]. In recent studies, a microenvironment was created using porcine dECM and endothelial cells via novel biofabrication methods. Three-dimensional printing and a dECM bioink enabled the fabrication of 3D pancreatic tissue constructs with enhanced functions [128]. Stem-cell-based insulin-producing cells for forming islet-like structures have been fabricated as biomimicking pancreatic islets [152]. Emerging evidence suggests that clustering insulin-producing cells inspired by pancreatic islet cells improve cell maturity and function [153,154]. To enable engineered pancreatic tissues to be transplanted, encapsulation strategies involving porous membranes or ECM-like hydrogels are typically adopted [155]. The cell encapsulation device can protect insulin-producing cells from the immune response, promote cell viability, and further advance successful transplantations [156,157].

Moderate fibrosis of the liver is a histological indicator of aging; however, some studies suggest that altered components in the ECM may initiate pathological progression as well [15]. Hepatic fibrosis, i.e., the accumulation of fibrillar ECMs, is a type of dyshomeostasis that results in end-stage liver disease [53]. Hepatic transplantation is an established treatment; however, to fill the gap between donor shortage and high demand, applied hepatocyte transplantation has been applied as an alternative solution in many cases [158]. As the microenvironment of the liver ECM is a critical cue for hepatocyte behaviors and functions, the liver dECM has been used in several studies [95,159,160,161,162]. The potential cell sources, as well as the MSCs and their derivatives of engineered hepatic tissues, were promoted during cell differentiation and hepatic-specific functions in the liver dECM hydrogel [161,162]. MSCs can differentiate into hepatocyte-like cells with therapeutic potential [163]. Biomimetic architecture is a necessary factor for the development of complex artificial tissues. The hollow fiber membrane, which mimics the layer of the actual liver, is a representative example [164]. A recent study used 3D bioprinting to construct the liver lobule architecture, thereby expanding the long-term function of hepatocytes and provided a physiologically relevant mechanical environment [159]. Another 3D bioprinting-based liver tissue engineering study verified that providing suitable biochemical and biomechanical surroundings enhanced the functions of HepG2 cells [95]. Furthermore, physiologically relevant hepatic models are being developed from advanced liver tissue engineering studies by building complex structures and applying co-culture systems [165]. A co-culture of hepatocytes, primary human sinusoidal endothelial cells, and stellate cells on a hollow fiber construct demonstrated self-reorganization and presented a tube-like structure that resembled a real liver tissue [164].

Vascular dysfunctions, which exhibit a higher risk with aging, can result in the decreased activity and deterioration of other organs. Regenerative strategies, including stem cell therapy, have been discussed [166]. Fabricating tubular-shaped tissues is challenging because maintaining a hollow structure with only cells and ECM materials is difficult. However, state-of-the-art biofabrication techniques have enabled the production of functional tubular tissues [167]. Diverse human blood vessel structures have been fabricated with multilevel and multibranch structures. The hollow channel was formed by removing a sacrificial material after the printing process [145]. Newly established coaxial bioprinting enabled the fabrication of hollow blood vessels, which can replace impaired blood vessels. A vessel dECM was mixed with alginate as a bioink and printed through a coaxial nozzle with a controllable, broad range of dimensions [136].

Retinal disorders, including macular degeneration and glaucoma, are age-related diseases. Additionally, cornea changes with age, including the thickening of both epithelial and endothelial basement membranes. As the cornea is known to be organized in a lattice pattern of collagen fibrils, which affords the transparency of the cornea, collagen rich biomaterials are used for inducing collagen fibril orientation. The corneal 3D models could be fabricated via drop-on-demand bioprinting using collagen-based bioink [168]. A recent study introduced a transparent, bioengineered corneal structure for transplantation. The engineered cornea was fabricated by inducing shear stress on a corneal stroma-derived decellularized extracellular matrix bioink based on a 3D cell-printing technique. The aligned collagen fibrils of dECM resulted in a highly mature and transparent corneal stroma analog. Epithelial and stromal cells were used for generating full-thickness corneas, and cell alignment was also investigated [133].

## 5. Future Perspectives and Conclusions

The ECM is vital to the human body. In addition to its structural role as a physical scaffold for cellular constituents, the ECM participates in numerous biological functions. Although the ECM is composed of non-cellular molecules, it undergoes significant changes during aging, including morphological changes that occur at the tissue level. Consequentially, tissues lose mass, dehydrate, undergo fibrosis, fail to regenerate, and cause dysfunctions. Age-related changes in the ECM are investigated in various forms. ECM-based biomaterials and tissue engineering approaches have been introduced to rejuvenate and regenerate aged tissues. ECM-based biomaterials, including the dECM, closely resemble the actual ECM and have tissue-specific compositions. ECM injection may aid tissue maintenance and compensate for volume losses caused by aging. Moreover, the ECM can be used as a carrier for cell delivery. The ECM provides a tissue-specific microenvironment for cells in vitro and in vivo.

Additionally, ECM-based materials can be used as cell delivery carriers and bioinks. SCs and their derivations can be encapsulated in ECM-based biomaterials and delivered to tissues with age-related dysfunction. The ECM supports the delivered cells, and the delivered cells successfully reconstruct the ECM, thereby validating its effect on restoring tissue function and regeneration. Furthermore, the ECM has been applied in tissue engineering. Engineered tissue can be fabricated using the ECM and cells using precise manufacturing methods such as 3D bioprinting. Fabricated organs, which are suitable for transplantation, can be substituted for donated organs. Biofabrication methods are currently being developed; with the progression of tissue engineering technology, every organ dysfunction associated with aging will soon be treatable.

The ECM demonstrated its versatility in various tissue engineering applications, and 3D bioprinting systems employ ECM-based materials as bioinks and biomaterial inks. The development of 3D bioprinting techniques has enabled the fabrication of complex engineered tissues. The potency of bioprinted tissues for regeneration strategies was verified to be efficient. Bioprinted tissues are used not only for rejuvenating aged tissues but also for treating injuries and chronic diseases. As biomimetics resemble real tissues, bioprinted tissues can be utilized as an in vitro test platform as well. Additionally, owing to the development of tissue engineering technology using ECM-based materials, transplanting without donors and performing in vivo tests without live animals can soon be realized. However, the properties of bioinks must be enhanced, and the accuracy of the manufacturing methods should be improved to fabricate more realistic tissues. Although ECM-based materials offer many advantages, some of their disadvantages must be addressed. They lack mechanical strength, are difficult to deposit precisely, and require a certain amount of time to crosslink. Therefore, further studies are required to improve their mechanical properties and address their weaknesses while maintaining their cell-friendly environment.

## Figures and Tables

**Figure 1 ijms-22-09367-f001:**
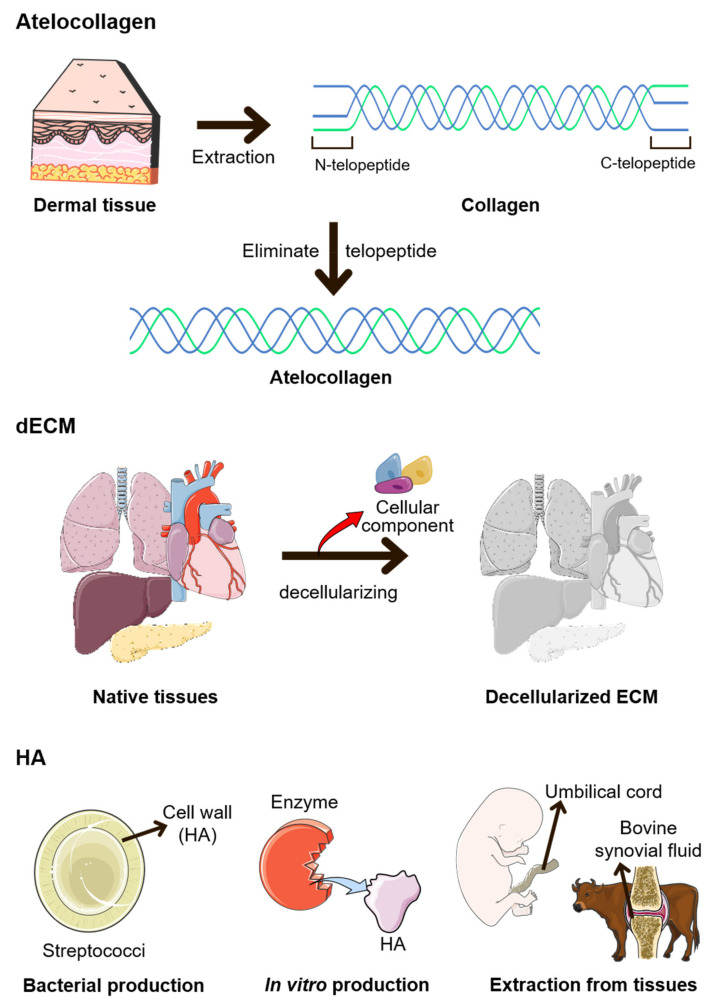
Methods for manufacturing modified ECMs. Atelocollagen is manufactured by removing telopeptides from collagen. Decellularized ECM (dECM) is manufactured by eliminating cellular components from native tissues. Finally, hyaluronic acid (HA) is produced using several methods. Bacteria such as *Streptococci* are representative bacterium that produce HA. Additionally, HA can be manufactured using enzymes, the umbilical cord, or bovine synovial fluid [83].

**Figure 2 ijms-22-09367-f002:**
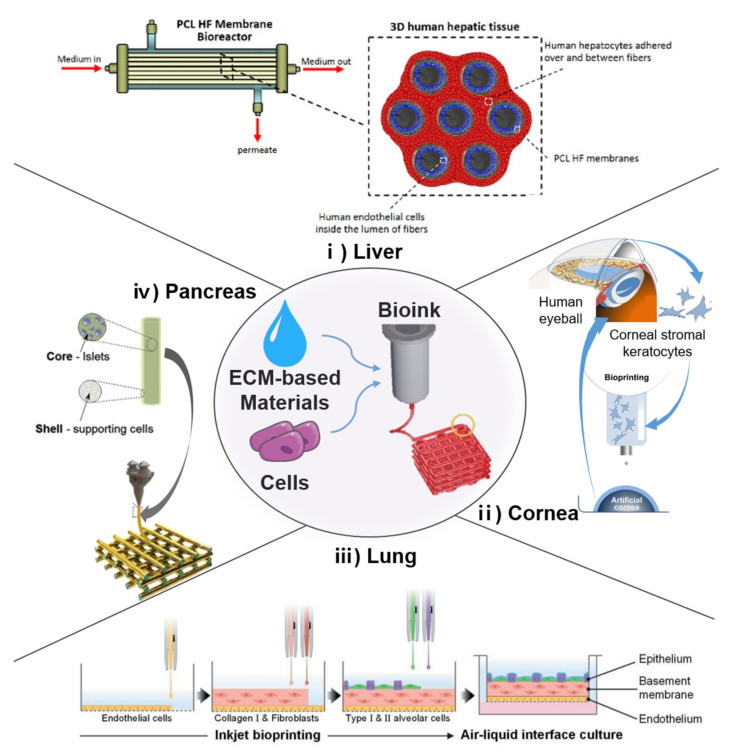
Applications of tissue engineering based on 3D bioprinting. (**i**) Poly(ε-caprolactone) (PCL) hollow fiber membrane bioreactor and scheme of 3D human hepatic tissue realized by culturing human hepatocytes over and between PCL HF membranes [129]. (**ii**) Bioprinting of functional and biomimetic 3D corneal model using hydrogels and cultivated human corneal stromal keratocytes [168]. (**iii**) Fabrication of alveolar barrier model with all cells inkjet-printed in a layer-by-layer manner [169]. (**iv**) coaxial printing approach for establishing vascularized bioartificial pancreatic constructs with pancreatic insulin-secreting cells housed in core component [170]. (clockwise).

**Table 1 ijms-22-09367-t001:** Features of senescence-related extracellular matrix (ECM) components.

Component	Functions and Properties	ECM-Aging Related Changes	
		Expression	Features
Collagen	- Provides structural properties and resilience to tissues [11] - More than 28 types of collagen exist [11]	↑	- Fibrosis [15] - Age-related neural disorders [16]
		↓	- Undergo modifications such as minimization and accumulation of advanced glycation end-products [18] - MMP-mediated degradation [18]
Fibronectin	- Binds to ECM molecules [19] - Interactions with the ECM network [20] - Mediates cellular activities by binding with receptors [20]	↑	- In aortic endothelial cells [22] - In patients with Alzheimer’s disease [16]
		↓	- In senescent hepatic stellate cells [21]
Elastin	- Provides mechanical strength, elasticity, firmness, and suppleness to tissues [24] - Exists in almost all connective tissues [24]	↓	- Destruction of elastin fiber structure [25,27] - Fragmentation and structural reorganization of elastin [27,28]
Laminin	- Major component of the basal lamina [29] - Aid mediating processes based on receptor- and matrix-binding properties [29]	↑	- In diabetic basement membrane [30]
		↓	- In an aging lung [31] - Non-progressive impaired neurotransmissions and premature morphological alterations [32]
Hyaluronic acid	- Water binding characteristics for tissue hydration [33] - Provides structure and organizations of the ECM [35]	↓	- Decline in the total mass and polymer size in aging skin [36] - Decrease in the water-binding capacity [37]

## Data Availability

Not applicable.

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
