# Peer review of "Employing Extracellular Matrix-Based Tissue Engineering Strategies for Age-Dependent Tissue Degenerations"

_ijms, 2021, doi:10.3390/ijms22179367_

Round 1

Reviewer 1 Report

Manuscript ID: ijms-1301139 describes the novel strategies based on ECM engineering to target aging dysfunctions. The authors have mentioned the implications of several ECM components and discussed the utilization of ECM for regeneration purposes.

Although the manuscript is interesting the following point should be addressed.

  • The title has some grammatical issues. Please correct.
  • The English language needs some polishing. There are some grammatical issues throughout the text.
  • Although the authors have mentioned the possibility of using decellularized ECM, current studies are suggesting complementary post-decellularization steps. This section has not been well developed.  Please refer to the following article for some examples especially for osteoarthritis applications. PMID: 33738088 
  • The sections on the recellularization of ECM are very poor. The authors could find many new research articles and clinical trials mostly based on the different types of MSCs. Please develop this part.
  • A very recent concept in tissue engineering and ECM remodeling modification is via using PRP. The authors are suggested to add some information about it.
  • In general, this is a very comprehensive review article, however, it remarkably lacks the focus on a specific subject. It seems that the authors are interested in elaborate different angles of ECM biology but it would have been better if they were focused more on cosmetic or dermal rejuvenation for example either than going from OA to cardiovascular disorder. 

Author Response

We would like to thank the reviewer for the valuable comments, which have helped improve our manuscript. We have carefully read the comments and revised the manuscript accordingly. Our point-by-point responses to these comments are shown in blue. The revisions that have been made in the manuscript are indicated in red. Please see the attachment.

Reviewer 2 Report

The paper is very interesting and deserves to be published after implementation.

Some references are redundant in my view, such as ref.1 as ECM should be known, other more innovative points lack references, such as the 3D-bioprinting, the ECM glycosylation (as ref. see https://www.future-science.com/doi/full/10.4155/fmc-2018-0368) and glycation ( as recent ref see doi: 10.14336/AD.2017.1121)

Some parts should be better organised for a better understanding.

In details, some suggestions:

pag 1, line 45. The ECM might be considered aging independent …….. This sentence could be misunderstood, I would suggest rephrasing to make immediately clear that ECM is aging dependent.

pag 2 line 58. A ref. such as a review on 3D extracellular matrix mimics could be helpful.

pag 3 line 101. Delate (messenger ribonucleic acid) mRNA is well known.

pag 3 line 105. Elastin is a fibrous self-assembly glycoprotein…. Glycosylation of elastin has not jet reported to my knowledge, there are some examples of glycation, a different process. I would suggest delating the term glycoprotein.

page 3 and 4, table 1. Even if full of information, the table is not “intuitive”. I would prefer a different organization, more essential and with arrows pointing up or down to indicate up or downregulation.

pag 4 line149. … affect cell ability. Which ability?

pag 5 line 200. add elastin

pag 5 line 212-216. I suggest to add the concept of glycation, being an ECM modification related to aging and diabetes.

pag 5 and 6 paragraph 4. The paragraph should be better structured, making clear that decellularized ECM can be used as entire tissue or in shredded form as component of bioinks for 3D printing/bioprinting. Furthermore, synthetic polymers are not all hydrophobic as stated in line 226. Ref 6o-62 seams not really inline with the topic.

pag 8 lines 288-289. The reported concept is questionable. Decellularization is performed in conditions (acids, bases, hypotonic and hypertonic solutions, detergents) that can generate loss of ECM components or changes in structures.

pag 9. The contribution of carbohydrates should be expanded (HA gradients, proteoglycans, ECM proteins N- and O-glycosylations)

Abbreviation, add PVA

Author Response

(The authors gave the same response as above.)

Round 2

Reviewer 1 Report

The authors have significantly improved their manuscript.

I have no further comments.